# An *in vitro* carcinogenesis model for cervical cancer harboring episomal form of HPV16

**Weerayut Wongjampa**[1,2], **Tomomi Nakahara**[3], **Katsuyuki Tanaka**[4], **Takashi Yugawa**[5], **Tipaya Ekalaksananan**[1,2], **Pilaiwan Kleebkaow**[2,6], **Naoki Goshima**[7], **Tohru Kiyono**[8]*, **Chamsai Pientong**[1,2]*

1 Faculty of Medicine, Department of Microbiology, Khon Kaen University, Khon Kaen, Thailand, 2 HPV & EBV and Carcinogenesis Research Group, Khon Kaen University, Khon Kaen, Thailand, 3 Department of Immune Medicine, National Cancer Center Research Institute, Tokyo, Japan, 4 Division of Carcinogenesis and Cancer Prevention, National Cancer Center Research Institute, Tokyo, Japan, 5 Department of Basic Medical Research, Division of Basic Medical Research, Japan Agency for Medical Research and Development, Tokyo, Japan, 6 Faculty of Medicine, Department of Obstetrics and Gynecology, Khon Kaen University, Khon Kaen, Thailand, 7 Department of Human Sciences, Musashino University, Tokyo, Japan, 8 Project for Prevention of HPV-Related Cancer, Exploratory Oncology Research and Clinical Trial Center, National Cancer Center, Chiba, Japan

* tkiyono@east.ncc.go.jp (TK); chapie@kku.ac.th (CP)

**Data Availability Statement:** All relevant data are within the paper and its Supporting Information files.

## Abstract

Deregulated expression of viral *E6* and *E7* genes often caused by viral genome integration of high-risk human papillomaviruses (HR-HPVs) into host DNA and additional host genetic alterations are thought to be required for the development of cervical cancer. However, approximately 15% of invasive cervical cancer specimens contain only episomal HPV genomes. In this study, we investigated the tumorigenic potential of human cervical keratinocytes harboring only the episomal form of HPV16 (HCK1T/16epi). We found that the HPV16 episomal form is sufficient for promoting cell proliferation and colony formation of parental HCK1T cells. Ectopic expression of host oncogenes, *MYC* and *PIK3CA^{E545K}*, enhanced clonogenic growth of both early- and late-passage HCK1T/16epi cells, but conferred tumor-initiating ability only to late-passage HCK1T/16epi cells. Interestingly, the expression levels of E6 and E7 were rather lower in late-passage than in early-passage cells. Moreover, additional introduction of a constitutively active *MEK1* (*MEK1DD)* and/or *KRAS^{G12V}* into HCK1T/16epi cells resulted in generation of highly potent tumor-initiating cells. Thus an *in vitro* model for progression of cervical neoplasia with episomal HPV16 was established. In the model, constitutively active mutation of PIK3CA, *PIK3CA^{E545K}*, and overexpression of *MYC*, in the cells with episomal HPV16 genome were not sufficient, but an additional event such as activation of the RAS-MEK pathway was required for progression to tumorigenicity.

## Introduction

Cervical cancer is one of the most common cancers among women worldwide. Persistent infection with high-risk human papillomaviruses (HR-HPVs) is a causal factor for cervical cancer development. Among all the HR-HPVs, HPV16 is the most prevalent, reaching 61% of

**Funding:** This study was supported by Japan Agency for Medical Research in the form of a grant to TK [22jk0210009], Khon Kaen University, Thailand in the form of a grant to PK [620008002], the Program Management Unit for Human Resources & Institutional Development, Research and Innovation in the form of a grant to TE [B05F630053], Research and Graduate Studies, Khon Kaen University in the form of a grant to CP [RP65-8-001], and by the Post-Doctoral Training Program from Khon Kaen University, Thailand to WW and CP [PD2562-12].

**Competing interests:** The authors have declared that no competing interests exist.

cervical cancer [1,2]. HPVs are non-enveloped, small DNA viruses with genomes of approximately 8,000 bps of double-stranded circular DNA. These viruses infect squamous epithelia at various anatomical sites including the cervix. HPV infection can cause hyper-proliferative lesions which are spontaneously resolved in most cases but can occasionally progress to cancer. In the productive viral life cycle, the viral genomes are established, maintained and amplified as nuclear episomes within keratinocytes. On the other hand, integration of the viral genomes is a crucial event which subverts the productive life cycle of the virus and is a major step towards carcinogenesis [3–5]. Typical integrants accompany complete or partial disruption of the E2 open reading frame (ORF) of the viral genome [6]. E2 protein functions as a transcriptional repressor for the viral early promoter driving expression of viral oncogenes, E6 and E7, thus the loss of E2's inhibitory function results in overexpression of E6 and E7. Furthermore, deregulation of the viral promoter by genetic or epigenetic changes and/or the formation of chimeric mRNA with cellular sequences are also hypothesized as the causes of E6 and E7 overexpression [7]. Up-regulation of the E6 and E7 oncogenes of HR-HPVs in basal epithelial cells promotes cervical carcinogenesis via their abilities to increase cell proliferation, allowing accumulation of chromosomal abnormalities. Thus, the integration of the HR-HPV genome into host DNA is regarded as a key, even prerequisite, step for the development of cervical cancer.

However, it is still debated whether integration is an early or late event in neoplastic progression of HPV-infected cells, since some invasive cervical cancer specimens contain only episomal genomes of HR-HPVs. For instance, various studies reported that only episomal genomes of HPV16 are detected in between 26% and 48% of cervical cancer specimens [8,9]. The analysis of HPV physical status in cervical samples using the amplification of papillomavirus oncogene transcripts (APOT) assay demonstrated HPV integration in 3% (5/172) of CIN 2 lesions, 17% (36/216) of CIN 3 lesions and 62% (95/153) of cervical carcinomas [10]. Therefore, HR-HPV genome integration can occur in early stages and increases during neoplastic progression [10]. However, in many cases of cervical cancer, HR-HPV genomes are present as episomes and often only as episomes without any integration. Interestingly, the frequency of integration may differ by HPV type. HPV16, HPV18 and HPV45 genomes were integrated in 55% (33/60), 92% (33/36) and 83% (20/24), respectively, of invasive cervical cancer samples. For HPV31 and HPV33 corresponding values were 14% (2/14) and 37% (7/19), respectively [10]. Integration of oncogenic HPV genomes in cervical lesions might be a consequence rather than the cause of chromosomal instability induced by deregulated high-risk *E6-E7* oncogene expression. To further investigate the significance of HPV integration, an experimental model mimicking cervical cancer with only episomal HPV genomes is needed.

Because *E6* and *E7* genes are always expressed in HPV-positive cervical cancer cells and can inactivate tumor suppressors, p53 and pRB, respectively, they are believed to play key roles in cervical carcinogenesis [11]. Epidemiological and experimental studies indicate that expression of E6 and E7 is necessary, but not sufficient, to induce cervical cancer and that additional genetic and/or epigenetic events are required [12]. Mutations or alterations in the expression of human oncogenes, including *MYC* [13], *PIK3CA* [14] and *RAS* [15], have been reported in cervical cancer. In previous studies, Narisawa-Saito et al. demonstrated that exogenous expression of oncogenic *MYC and HRAS*$^{G12V}$ together with HPV16 E6E7 is sufficient for tumorigenic transformation of normal human cervical keratinocytes (HCKs) [16]. However, the effect of host oncogenes in cervical cancer harboring the episomal form of HPVs has not been extensively examined. The proto-oncogene *MYC* encodes a transcription factor that regulates cell proliferation, growth and apoptosis [17,18]. Deregulated expression or function of MYC is one of the most common abnormalities in human malignancy including cervical cancer [13,19]. The phosphatidylinositol 3-kinases (PI3Ks) are lipid kinases that regulate signaling pathways important for neoplasia, including cell proliferation, adhesion, survival, and motility

[20]. Amplification and/or somatic mutations within the alpha catalytic subunit of PI3K *(PIK3CA)* are present in many human cancers including cervical cancer [21]. In cervical cancer patients with *PIK3CA* mutations, approximately 60% contain the E545K substitution, a mutation associated with an enhanced migratory phenotype in cervical cancer cells [22]. Thus, mutation or alteration in the expression of *MYC* and *PIK3CA* is frequently associated with development of cervical cancer.

In this study, we aimed to investigate the tumorigenic transformation of HCK1T cells carrying episomal HPV16 genomes (HCK1T-HPV16). In previous studies, the tumor cell line, TC-1, was generated from primary epithelial cells of C57BL/6 mice. The cells were immortalized with HPV16 E6 and E7 and subsequently transformed with HPV16 E6 and E7 and human c-Ha-ras oncogenes [23]. In addition, we have previously demonstrated that activated RAS and MYC overexpression in combination with HPV16 E6E7 overexpression in primary cervical keratinocytes confers high tumorigenicity to the cells. Therefore, in this study we examined the tumorigenic potentials of HCK1T-HPV16 with ectopic expression of oncogenic *MYC* and $PIK3CA^{E545K}$. Furthermore, we tested whether additional alteration, such as of the genes *MEK1DD* or $KRAS^{G12V}$, is required for tumorigenesis and/or further progression. To our knowledge, this is the first *in vitro* carcinogenesis model for cervical cancer harboring HPV16 episomes without any sign of integration. Our results indicate what might be the minimum requirement of specific oncogenes for the development of cervical cancer harboring episomal forms of HR-HPV. These findings provide a better understanding for the roles of cellular and viral oncogenes in cervical carcinogenesis with or without HPV16 integration.

## Materials and methods

### Cell culture

Establishment of HCK1T and HCK1T-HPV16 cells was described previously [24,25]. Briefly, HCK1T was co-transfected with pAd/HPV16/neo which encodes the complete HPV16 genome flanked by two loxP sites, and with pxCANCre vector expressing Cre recombinase followed by G418 selection to establish HCK1T-HPV16 cells. These cells were maintained in Epi-Life medium (Life Technologies, Grand Island, NY) supplemented with G418, penicillin and streptomycin (Sigma-Aldrich, St. Louis, MO, USA). The cell line was authenticated by the existence of HPV16 genome and expression of viral genes, E6 and E7, by western blotting as indicated in the Figures of this paper.

HeLa and 293T cells were cultured in Dulbecco's modified Eagle's medium (DMEM) supplemented with 10% fetal bovine serum (FBS) and antibiotics, penicillin and streptomycin (Sigma-Aldrich, St. Louis, MO, USA). All cells were cultured at 37°C in a 5% $CO_2$ incubator. Doxycycline (DOX) and 4-hydroxytamoxifen (4-OHT) were dissolved in 70% ethanol and stored as high-concentration stocks at 4°C until used at the indicated concentrations. The concentrations of the stock solutions were as follows: DOX (Clontech, Mountain View, CA, USA) at 1 mg/ml and 4-OHT (Sigma-Aldrich, St. Louis, MO, USA) at 100 μM. The cervical cancer cell lines, HeLa (a gift from Dr. M. Inagaki, JCRB9004, ICRB Cell Bank in 1996) were authenticated by Short Tandem Repeat (STR) analysis, expression of HPV18 E7 was confirmed by immunoblotting in September 2012. 293T cells were just used for virus packaging and the high transfection efficiency was confirmed by EGFP expression in each transfection and expression of SV40 large T antigen was confirmed in September 2012.

### Plasmid construction, cell transfection and transduction

Expression plasmids were constructed by using the Gateway system according to the manufacturer's instructions (Invitrogen, Life Technologies, Saint Aubin, France). An entry vector

encoding human *PIK3CA^L30M* (clone FLJ75190AAAN, NITE Biological Resource Center, Japan) was subjected to site-directed mutagenesis to construct an entry vector encoding *PIK3-CA^E545K*, pENTR221-PIK3CA^E545K. An N-terminal 3XFLAG tag was inserted by inverse PCR to PIK3CA^E545K and then F2A peptide sequence was inserted at 5' of 3XFLAG-PIK3CA^E545K by in-fusion reaction (Clontech, Palo Alto, CA, USA) to generate pENTR221-F2A-3XFLAG-PIK3CA^E545K. pENTR221-MYC-F2A1-3xFLAG-PIK3CA^E545K was constructed by in-fusion reaction, inserting a *MYC* fragment from CSII-TRE-Tight-MYC-F2A1-HRASG12V [26] at 5' of F2A and then recombined with an expression vector, PB-TAC-ERN (kindly provided by Knut Woltjen in CiRA) to generate PB-TAC-ERN-MYC-F2A1-3xFLAG-PIK3CA^E545K by LR reaction. A lentivirus vector expressing HA-tagged MEK1DD or MYC with tetracycline-inducible system, CSII-TRE-Tight-HA-MEK1DD or CSII-TRE-Tight-MYC, respectively and a retrovirus vector pQCXIP-mERT2-KRAS^G12V were previously described [27]. PiggyBAC transposase gene was inserted into pCDH-EF-RfA-IRES-puro, derived from pCDH-EF1α-MCS-IRES-Puro (System Biosciences LLC, Palo Alto, CA) by LR reaction. Detailed methods for the construction of the plasmids are available upon request. The production of recombinant lenti- and retro-viruses has been described previously [28]. PB-TAC-ERN-MYC-F2A1-3xFLAG-PIK3CA^E545K and pCDH-EF-transposase-IRES-puro were co-transfected at the molar ratio of 1:1 to HCK1T-HPV16 cells using FuGENE HD (FUGENT LLC, Madison, WI) followed by sequential drug selection, first puromycin (1 μg/ml) for 2 days and then G418 (50 μg/ml) for 7 days so as to generate HCK1T-HPV16-tetON-MYC-F2A-3xFLAG-PIK3-CA^E545K cells. As the expression levels of MYC in HCK1T-HPV16-tetON-MYC-F2A-3xFLAG-PIK3CA^E545K cells was unexpectedly low possibly due to inefficient cleavage of F2A peptide, they were subjected to serial lentivirus-mediated gene transduction with CSII-TRE-Tight-MYC at MOI = 10. The resultant cells were further transduced with CSII-TRE-Tight-HA-MEK1DD at MOI = 10 without drug selection or pQCXIP-mERT-KRAS^G12V at MOI = 3 followed by selection with puromycin (1 μg/ml). All the cell lines were used as pooled population without cloning.

## Viral copy number detection by quantitative PCR

Genomic DNA (gDNA) was isolated from monolayer cultures of cells and tumor tissues using Wizard SV Genomic DNA Purification System according to the manufacturer's instructions (Promega, Madison, WI, USA). The isolated gDNA was quantified by spectrophotometry. Fifty nanograms of the gDNA was subjected to quantitative PCR (qPCR) using StepOne plus (Applied Biosystems, Foster City, CA) as previously described [24]. Serial dilutions of linearized HPV16 genome purified from pUC-HPV16 plasmid DNA digested with BamHI were used as standards to measure the amount of HPV16 DNA. The copy number of HPV16 per cell was estimated based on the assumption that total human genomic DNA is 6.6 pg/diploid cell. PCR primers were designed within the *E6* and *L2* ORFs of HPV16. All PCRs were run in triplicate. P-values were determined using Student's t-test. The following primers were used. E6: Forward 5'–GAACTGCAATGTTTCAGGACCC-3' and Reverse 5'–TGTATAGTTGTTTGCAGCTCTG TGC-3', L2: Forward 5'–ACAGATACACTTGCTCCTGTAAGACC-3' and Reverse 5'– GCAGGTGTGGTATCAGTTGAAGTAGT-3'.

## RNA extraction and reverse transcription (RT)-qPCR

Total RNA was isolated using a RNeasy Plus Mini kit (Qiagen, Venlo, Netherlands) following the manufacturer's instructions. One microgram of total RNA was subjected to 10 μl reverse transcription (RT) reaction using a PrimeScript RT reagent kit (TAKARA BIO INC, Shiga, Japan) and 1 μl of the RT products was used for real-time PCR reactions to measure the

mRNAs of interest. The PCR mixtures were prepared using KAPA SYBR FAST qPCR kits (Kapa Biosystems, Wilmington, MA, USA) and the real-time PCR was performed with StepOnePlus (Applied Biosystems, Foster City, CA). The relative levels of the target mRNAs were calculated by a ΔΔCT method using GAPDH mRNA as an internal control. All PCRs were run in triplicate. P-values were determined by Student's t- test. The following primers were used. GAPDH: Forward 5′-TCATCAGCAATGCCTCCTGCA-3′ and Reverse 5′-TGGGTGG CAGTGATGGCA-3′, HPV16 E2: Forward 5′-GAACTGCAACTAACGTTAGA-3′ and Reverse 5′-TCCATCAAACTGCACTTCCA-3′, HPV16 E6: Forward 5′-GAACTGCAATGTT TCAGGACCC-3′ and Reverse 5′-TGTATAGTTGTTTGCAGCTCTGTGC-3′, HPV16 E7: Forward 5′-GGAGGAGGATGAAATAGATGGTC-3′ and Reverse 5′-AGTACGAATGTCTA CGTGTGTGC-3′.

## Clonogenic assay

The clonogenic assay was performed as previously described [29]. Briefly, cells were harvested from exponential phase cultures by trypsinization, counted and seeded at 500 cells per well into six-well plates (Falcon, Corning, NY, USA) containing 2 mL of EpiLife medium (Life Technologies, Grand Island, NY) supplemented with antibiotics, penicillin and streptomycin (Sigma-Aldrich, St. Louis, MO, USA). Thereafter, the plates were incubated in a 5% $CO_2$ incubator at 37°C without refeeding for 2 weeks. After incubation, the medium was aspirated and cells were washed with PBS. Colonies were fixed with fixation solution (1:7 v/v acetic acid/ methanol) at room temperature (RT) for 5 minutes, then stained with Giemsa's dye at RT for 2 hours. Colonies were washed with tap water and counted under a microscope.

## Western blot analysis

Whole-cell proteins were extracted in a lysis buffer (50 mM Tris-HCl, 250 mM NaCl, 5 mM EDTA, 1% NP-40, 20% glycerol, 0.1% SDS, 1% Deoxycholate) supplemented with 5% (v/v) protease inhibitor cocktail (Nacalai Tesque, Kyoto, Japan) and phosphatase inhibitors (500 μM sodium orthovanadate, 100 mM sodium fluoride, 10 mM sodium pyrophosphate). SDS-polyacrylamide gels were loaded with 20 μg of total cell lysate per lane as described previously [25]. A monoclonal antibody to HPV16 E6 (clone 47A4) raised against the N-terminal 16 amino acids peptide of HPV16 E6 was used for detection of the E6 protein [30]. All other antibodies were purchased as follows: primary antibodies against HPV16 E7 (clone 8C9) (Invitrogen, Carlsbad, CA, USA), p53 (clone Ab6) (Oncogene Science/EMD Millipore, Billerica, MA, USA), MYC (clone C33), KRAS (clone F234) (Santa Cruz Biotechnology, Santa Cruz, CA, USA), PIK3CAp110 (cat no. 4249), p-AKT (cat no. 9271), AKT (cat no. 9272), p-mTOR (cat no. 2971), mTOR (cat no. 2972), MEK1/2 (cat no. 8727), p-ERK (cat no. 9101), ERK (cat no. 9102) (Cell Signaling Technologies, Danvers, MA, USA) and vinculin (cat no. v9264) (Sigma-Aldrich, St. Louis, MO, USA). Horseradish peroxidase-conjugated anti-mouse or anti-rabbit (Jackson Immunoresearch Laboratories, West Grove, PA, USA) immunoglobulins were used as secondary antibodies. The LAS3000 CCD-Imaging System (Fujifilm Co. Ltd, Tokyo, Japan) was employed for detection of proteins visualized by Lumi-light plus western blotting substrate (Roche Applied Science, Penzberg, Germany).

## Mouse xenograft

All surgical procedures and care administered to the animals were in accordance with institutional guidelines of the National Cancer Center in Japan. Prior to implantation, cells were incubated with 1 μg/ml DOX and/or 100 nM 4-OHT to induce expression of designated oncogenes for 2 days. A 100 μL volume of cell suspension mixed with Matrigel (BD Biosciences,

San Jose, CA, USA) at 1:1 was subcutaneously injected into female BALB/c nude mice (Clea Japan Inc., Tokyo, Japan) at $1\times10^6$ cells per site, under isoflurane anesthesia (20% v/v isoflurane in propylene glycol) in the clean bench. DOX and 4-OHT were administered to mice via drinking water at concentrations of 1 mg/ml and 0.2 mg/ml, respectively. The mice were sacrificed by deep isoflurane anesthesia, then the tumors were collected.

### Statistical analysis

The data are presented as the mean ± SEM (standard error of the mean) and were compared using Student's t-test or one-way ANOVA tests, as appropriate, in the Statistical Program for Social Sciences 13.0 software (SPSS Inc., Chicago, IL, USA). Results were considered statistically significant at $P < 0.05$. All graphs were drafted using GraphPad Prism version 5.00.286 (GraphPad Prism, San Diego, California, USA).

## Results

### Characterization of HCK1T cells harboring episomal form of HPV16 genome

We started to establish an HPV16 episome-mediated carcinogenesis model of cervical cancer with HCK1T-HPV16, HCK1T stably maintaining episomal HPV16 genomes which was experimentally generated in our laboratory [24,25]. In this study, we used the term HCK1T/16epi to emphasize episomal HPV16 genomes. As described previously, HCK1T/16epi cells contained only episomal HPV16 genomes with approximately 100 copies per cell [24,25] though subpopulation of the cells derived from the same origin harbored approximately 50–100 copies per cell compared to standard plasmid (S1 Fig). The viral copy numbers in HCK1T/16epi were consistent with the copy numbers reported in clinical samples. Morphology of HCK1T/16epi cells at early passage (p29, 9 passages after introduction of episomal HPV16 [24,25]) and late passage (p51, 31 passages after transduction of episomal HPV16 [24,25]) resembled that of parental HCK1T (Fig 1A). The copy numbers of episomal HPV16 genomes were consistent in the cells isolated from early and late passage (Fig 1B). Since HCK1T/16epi cells were passaged every 5–7 days, these results demonstrated that cell morphology and the viral copy numbers are stably maintained over at least 110 days or 50 population doublings. Transcription levels of viral early genes, E2, E6 and E7, in HCK1T/16epi were also compared between early- and late-passage cells. As shown in Fig 1C and 1D, E2, E6 and E7 mRNA levels were higher in early-passage cells than they were in late-passage cells. In collinear with mRNA levels, western blot analysis showed that the expression levels of E6 and E7 oncoproteins were higher in early-passage cells than that in late-passage cells (Fig 1E). Cell proliferation and clonogenic assays showed a significant increase in cell proliferation and colony formation of HCK1T/16epi compared with parental HCK1T. Interestingly, late-passage HCK1T/16epi showed significantly higher ability to form colonies compared with the early-passage cells, even though the expression levels of viral oncoproteins, E6 and E7, were lower (Fig 1F and 1G). Consistent with previous results, these cells failed to form tumors in nude mice without additional alterations [16,26]. Our results indicated that the presence of the HPV16 genome in episomal form can enhance proliferation and clonogenic potential of HCK1T cells but was insufficient to induce tumorigenic transformation.

### Sequential transduction of oncogenes into HCK1T cells harboring episomal form of HPV16 genome

HCK1T/16epi cells at early and late passage were further transduced with *MYC* and *PIK3-CA$^{E545K}$* oncogenes using the Tet-On expression vector system as described in Materials and

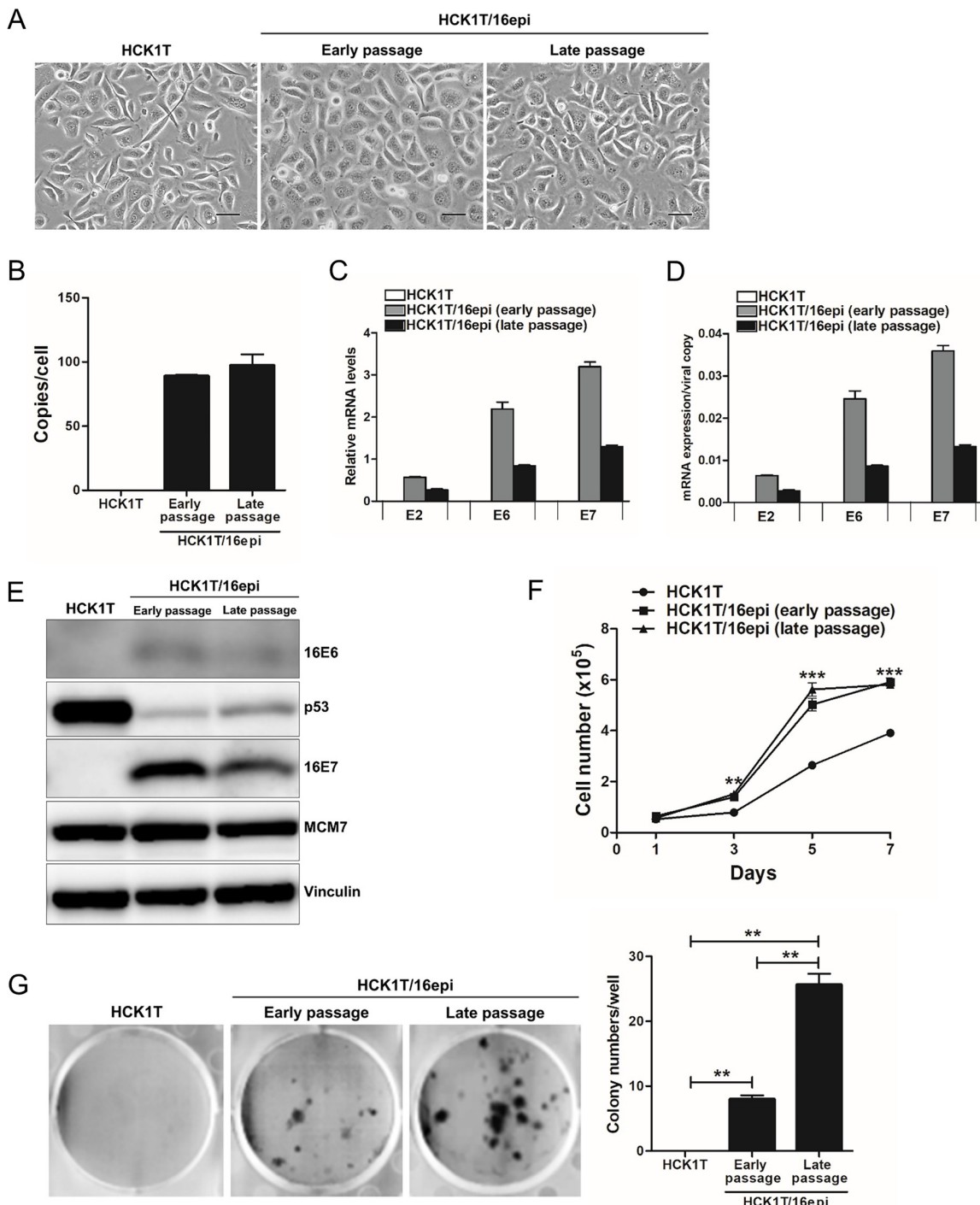

**Fig 1. Characterization of the HCK1T cells harboring episomal form of HPV16 genome.** (**A**) Cell morphology of HCK1T/HPV16epi at early and late passage was compared to parental HCK1T cells. Scale bars represent 50 μm. (**B**) The copy number of HPV16 was measured by quantitative PCR using specific primers to the E6 and L2 ORFs of HPV16. Relative expression levels of HPV16 E6 and E7 mRNA in HCK1T/16epi, showing levels overall (**C**) and per viral copy, calculated using the following formula: Mean relative mRNA expression/mean relative HPV16 copy number (**D**). (**E**) Expression of HPV16 E6 and E7 oncoproteins was detected by western blotting using specific anti-HPV16 E6 and E7 antibodies and the levels of E6 and E7 were compared between early- and late-passage HCK1T/16epi cells. Parental HCK1T cells were included as a negative control. Levels of p53 and MCM7 proteins were also determined to show that E6 and E7 from episomal HPV16 genomes are able to reduce p53 or increase MCM7, a robust indicator of E7 activity [31], respectively. Vinculin was included as a loading control. The ability of the episomal form of HPV16 to promote cell transformation in HCK1T/16epi cells relative to the parental HCK1T cell line was demonstrated by examining growth curves and clonogenic potential. (**F**)

Growth curves of early- as well as late-passage HCK1T/16epi cells and parental HCK1T cells. (**G**) Clonogenic potential of HCK1T/16epi and parental HCK1T cells. Each bar represents the mean of triplicate values ± SEM. **$P \leq 0.01$, ***$P \leq 0.001$.

Methods. No overt change in morphology was detected when the expression of specific oncogenes was induced with DOX (Fig 2A). The expression of individual transgenes and their downstream target proteins was confirmed by western blotting. As shown in Fig 2B, MYC and PIK3CAp110 protein levels were increased in the presence of DOX. Increased phosphorylation of AKT, a downstream target of PIK3CA, indicated that the expression of exogenous PIK3-CA$^{E545K}$ augments activation of AKT. E6 protein expression was detected at low levels when compared with E6 gene expression level in Fig 1C and the protein band is unclear. This problem might be effected from the antibody clone used. However, E6 gene expression can be confirmed by the level of mRNA detected in the cells, shown in Fig 1C. The tumor-promoting potential of early- and late-passage HCK1T/16epi expressing the *MYC* and *PIK3CA$^{E545K}$* oncogenes was examined by generating mouse xenografts. In this experiment, two mice were entirely used and four injection sites were set per mouse. As shown in Fig 2C and 2D, overexpression of *MYC* and *PIK3CA$^{E545K}$* was sufficient to confer tumorigenicity to late-passage but not early-passage HCK1T/16epi cells. The presence and copy number of episomal HPV16 DNA in tumors generated with the late-passage HCK1T/16epi expressing *MYC* and *PIK3-CA$^{E545K}$* was examined by qPCR with two independent set of primers targeting E6 ORF and L2 ORF because total genomic DNA recovered from the tumors was not sufficient for southern blot analysis (S2 Fig). Pathological features of squamous cell carcinoma which are poorly differentiated malignant cells were apparent by hematoxylin and eosin (H&E) staining (Fig 2D). From these data, we conclude that expression of *MYC* and *PIK3CA$^{E545K}$* can readily confer a tumorigenic phenotype to only late-passages HCK1T/16epi cells. We next searched for additional alterations that might promote full transformation of early-passage HCK1T/16epi cells. As we reported that expression of HPV16 E6E7 and KRAS$^{G12V}$ was sufficient to induce tumorigenicity of primary HCK cells, and the major signaling pathways downstream of KRAS include the PIK3CA/AKT and the MEK/ERK pathways, we focused on these two pathways, Both early- and late-passage HCK1T/16epi cells expressing tet-inducible *MYC* and *PIK3-CA$^{E545K}$* were further transduced with DOX-inducible expression of constitutively active *MEK1*, *MEK1DD*, or *mERT-KRAS$^{G12V}$* (*ER-KRAS*). In these cells, expression of *MYC*, *PIK3-CA$^{E545K}$* and *MEK1DD* could be induced by addition of DOX, and function of *ER-KRAS$^{G12V}$* chimeric protein could be activated by addition of 4-OHT. No overt changes in morphology were observed in HCK1T/16epi cells when expression of these oncogenes was induced (Fig 3A and 3B). The episomal status and the copy numbers of HPV16 DNA were not significantly different between the presence and the absence of the drugs (DOX and 4-OHT) in both early- and late-passage cells (Figs 3C, 3D and S1). The expression of individual transgenes and their downstream target proteins was confirmed by western blotting. As shown in Fig 3E and 3F, the expression of MYC, PIK3CAp110, MEK and KRAS proteins increased in the present of DOX and/or 4-OHT. Increased phosphorylation of AKT and mTOR which are the downstream target of PIK3CAp100 and also ERK which is the downstream target of KRAS and MEK confirmed that induced transgenes are functionally active. The expression levels of E6 and E7 proteins were comparable among parental HCK1T/16epi. Interestingly, the expression levels of E7 were decreased in *KRAS$^{G12V}$*-transduced cells compared to other cells. In addition, the HCK1T/16epi cells at late passage transduced with oncogenes were examined for their clonogenic potential. As expected, ectopic expression of *MYC/PIK3CA$^{E545K}$* irrespective of additional expression of *MEK1DD* or *ER-KRAS$^{G12V}$* enhanced clonogenic ability. However,

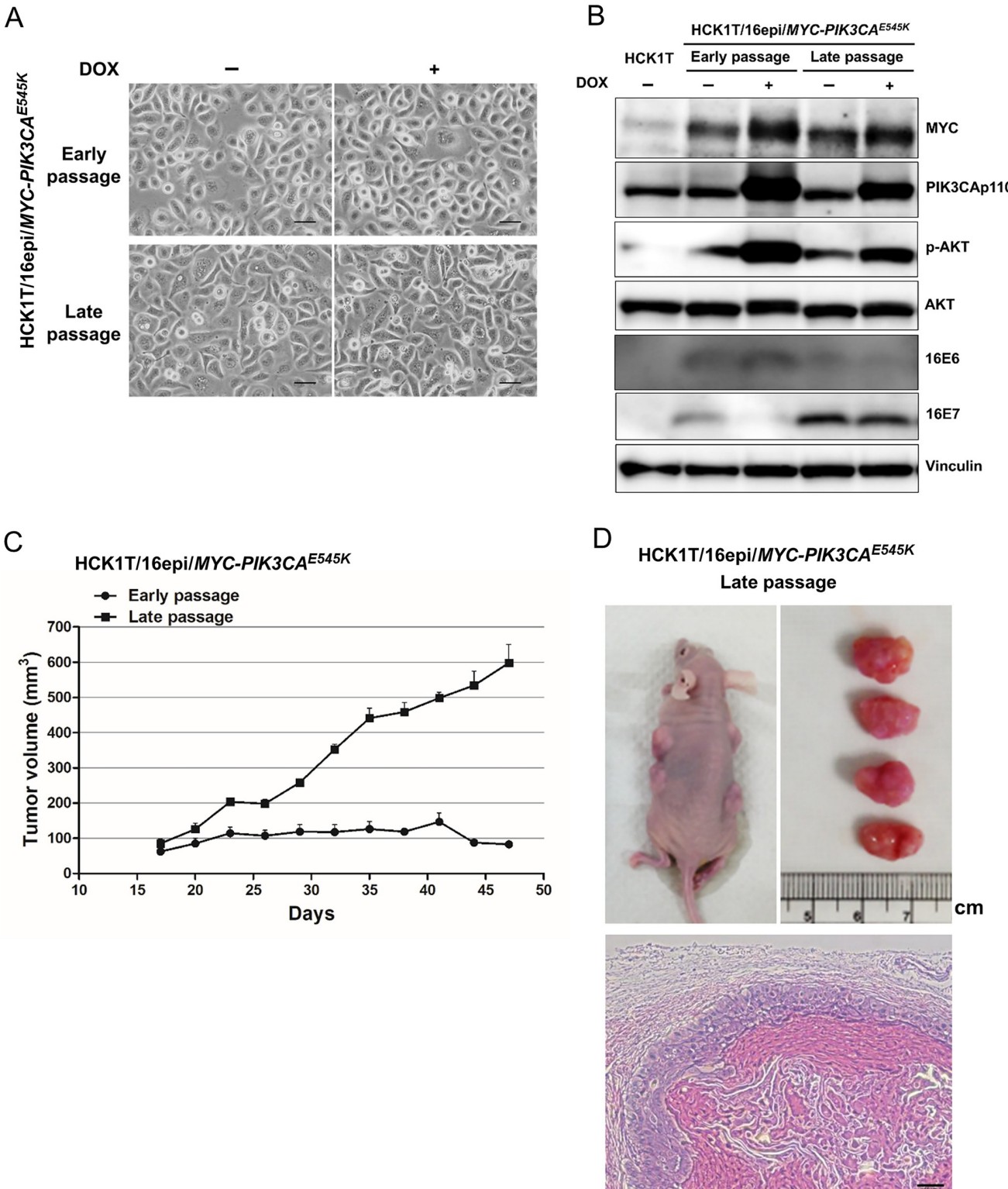

**Fig 2. Establishment of an *in vitro* model for cervical cancer with cells harboring episomal form of HPV16 genome and expression of defined oncogenes.** HCK1T/16epi cells transduced with indicated oncogenes were incubated with or without 1 μg/ml doxycycline (DOX) for 4 days. (**A**) Cell morphology of HCK1T/16epi cells did not change after expression of the indicated oncogenes. Scale bars represent 50 μm. (**B**) The expression of individual transgenes including MYC and PIK3CAp110 and their downstream target proteins including AKT and p-AKT were detected by western blotting. The expression levels of HPV16 E6 and E7 proteins in HCK1T/16epi cells were also examined. Vinculin was included as a loading control. (**C**)

Tumor-promoting potential of the *MYC* and *PIK3CA*$^{E545K}$ oncogenes were compared between early and late passage of HCK1T/16epi cells by mouse xenografts. 1x10$^6$ cells mixed with Matrigel were subcutaneously injected into nude mice. Mice were given 1 mg/ml DOX. The size of each tumor was measured at the indicated time points. (**D**) Mice were sacrificed after tumor formation then tumor mass was measured. H&E staining of representative tumors isolated from nude mice injected with HCK1T/16epi expressing *MYC* and *PIK3CA*$^{E545K}$. Scale bars represent 50 μm.

additional expression of MEK1DD or activation of *ER-KRAS*$^{G12V}$ did not further enhance clonogenicity (S3 Fig). Activation of *ER-KRAS*$^{G12V}$ alone in HCK1T/16epi cells induced macropinocytic cell death as reported previously [27].

## Tumorigenicity in nude mice

Early- and late-passage HCK1T/16epi cells transduced with additional oncogenes were pretreated with DOX and/or 4-OHT or vehicle for 2 days and then subcutaneously transplanted into female BALB/c nude mice. In this experiment, twelve mice were entirely used and four injection sites were set per mouse. After cells injection, mice were then given 1 mg/ml DOX and/or 0.2 mg/ml 4-OHT, or vehicle via drinking water. As shown in Fig 4A and 4B, HCK1T/16epi with expression of both *MYC/PIK3CA*$^{E545K}$ and *MEK1DD* resulted rapid tumor formation regardless of the early-passage or the late-passage; tumor volumes reached more than 600 mm$^3$ in all mice (100%; 4 of 4 for each passage) within 4 weeks for early-passage and 3 weeks for late-passage cells. HCK1T/16epi with the expression of *MYC/PIK3CA*$^{E545K}$ and activation of *ER-KRAS*$^{G12V}$ also formed tumors (100%; 4 of 4) within 6 weeks for early-passage and 5 weeks for late-passage cells. We noted that early-passage HCK1T/16epi cells overexpressing *MYC* and *PIK3CA*$^{E545K}$ give rise tumors in 2 out of 3 mice (Fig 4C) in this particular experiment, whereas late-passage cells gave rise to tumors in all of 3 mice (Fig 4D). The differences in results of early-passage cells may be due to the differences in the drinking behavior of the mice, resulting in different doses of DOX uptake and leading to different tumor formations. Consistent with earlier observations, HCK1T/16epi cells expressing only *MYC/PIK3CA*$^{E545K}$ formed tumors at a much slower rate (100%; 4 of 4; within 2.5 months for early-passage and 2 months for late-passage cells). In contrast, HCK1T/16epi cells without induction of oncogenes failed to form tumors in nude mice untreated with DOX and/or 4-OHT (Fig 4A and 4B). In addition, the tumor volumes decreased after discontinuation of DOX administration, indicating the dependence of the tumor growth on the expression of *MYC* and *PIK3CA*$^{E545K}$ (Fig 4B). The presence and copy number of episomal HPV16 DNA in tumors generated with the early- and late-passage cells were examined by qPCR with two independent set of primers targeting E6 ORF and L2 ORF. Interestingly, the viral copy number in early passage cells was decreased in oncogenes-transduced cells compared to late passage cells (S4 Fig). Pathological features typical of squamous cell carcinoma which are poorly differentiated malignant cells were confirmed by H&E staining of these tumors (Fig 5A and 5B). We conclude that introduction of *MEK1DD* or *ER-KRAS*$^{G12V}$ in addition to expression of *MYC* and *PIK3CA*$^{E545K}$ confer tumor initiating ability to early-passage HCK1T/16epi cells.

## Discussion

The molecular etiology of cervical cancer in which the episomal form of HR-HPV presented is poorly understood due to the lack of a well-defined model. Therefore, we aimed to establish an *in vitro* model to allow the reconstruction of events leading to HR-HPV episome-mediated carcinogenesis. In this model, we used HCK1T cells harboring episomal form of the HPV16 genome (HCK1T/16epi) of relatively early and late passage. Interestingly, late-passage HCK1T/16epi showed higher clonogenic ability than the early-passage of the same cells did. Similarly, Wechsler et al. have reported that HSIL-like lines of normal immortalized human

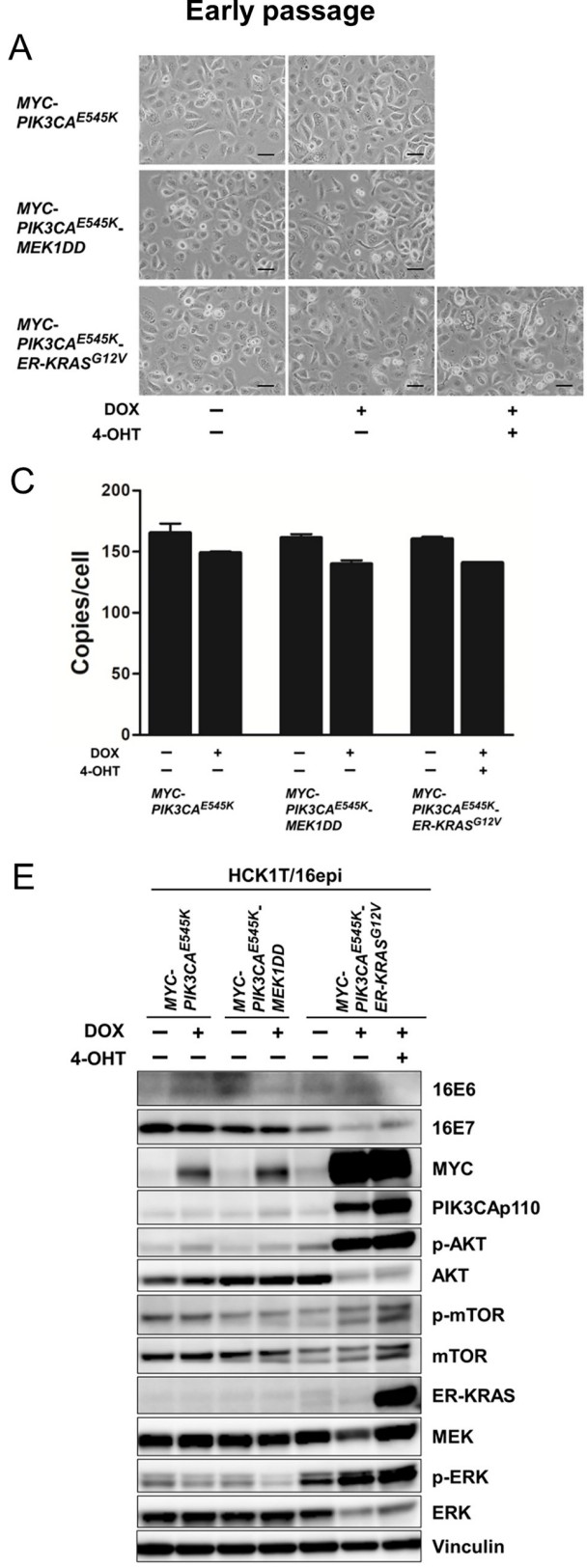

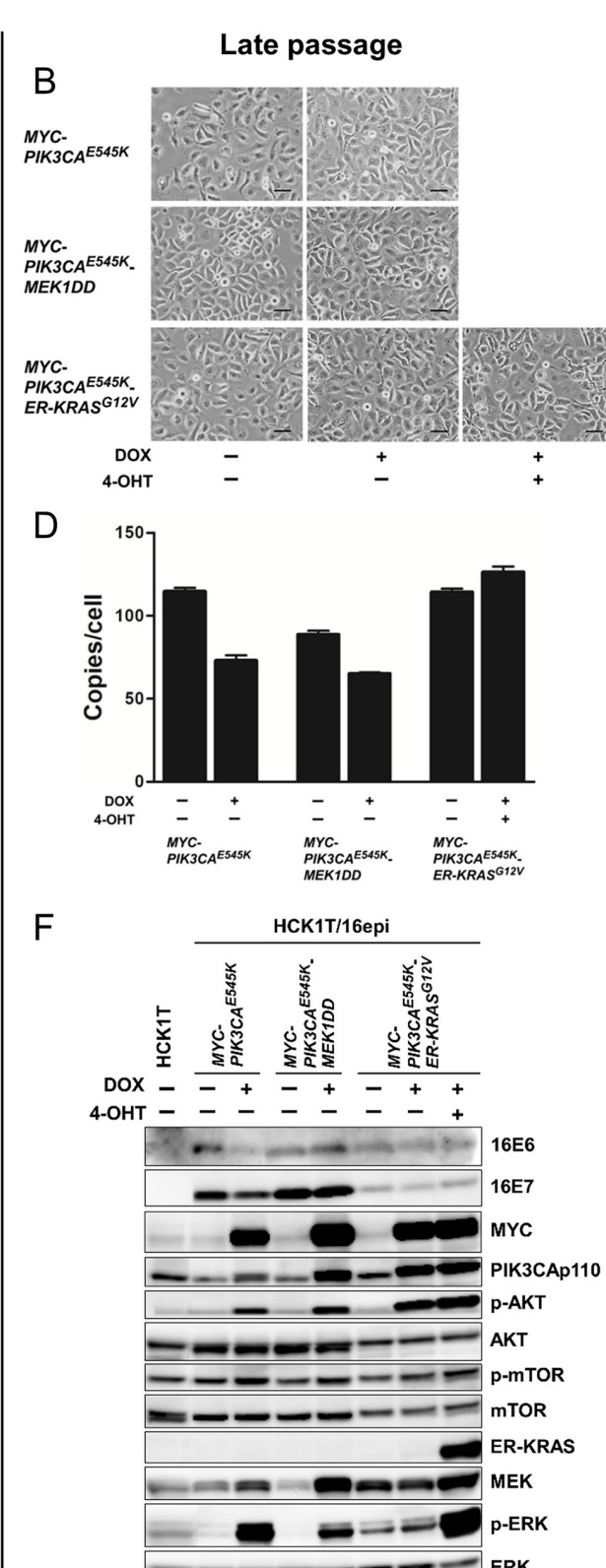

**Fig 3. Characterization of the HCK1T/16epi cells expressing human oncogenes.** HCK1T/16epi cells transduced with the indicated oncogenes were incubated with or without 1 µg/ml DOX and/or 100 nM 4-OHT for 4 days. Cell morphology of HCK1T/16epi cells from early passage (**A**) as well as late passage (**B**) did not change after expression of the indicated oncogenes. Scale bars represent 50 µm. The copy number of HPV16 genomes was determined by quantitative PCR using primers specific to the *E6* and *L2* ORFs of HPV16, at early passage (**C**) and at late passage (**D**). Each bar represents the mean of triplicate values ± SEM. The expression of individual transgenes including MYC, PIK3CAp110, MEK and KRAS, and their downstream target proteins including AKT; p-AKT, mTOR; p-mTOR and ERK; p-ERK were detected by western blotting, in the cells from early passage (**E**) and late passage (**F**). The expression levels of HPV16 E6 and E7 proteins in HCK1T/16epi cells were also examined. Vinculin was included as a loading control.

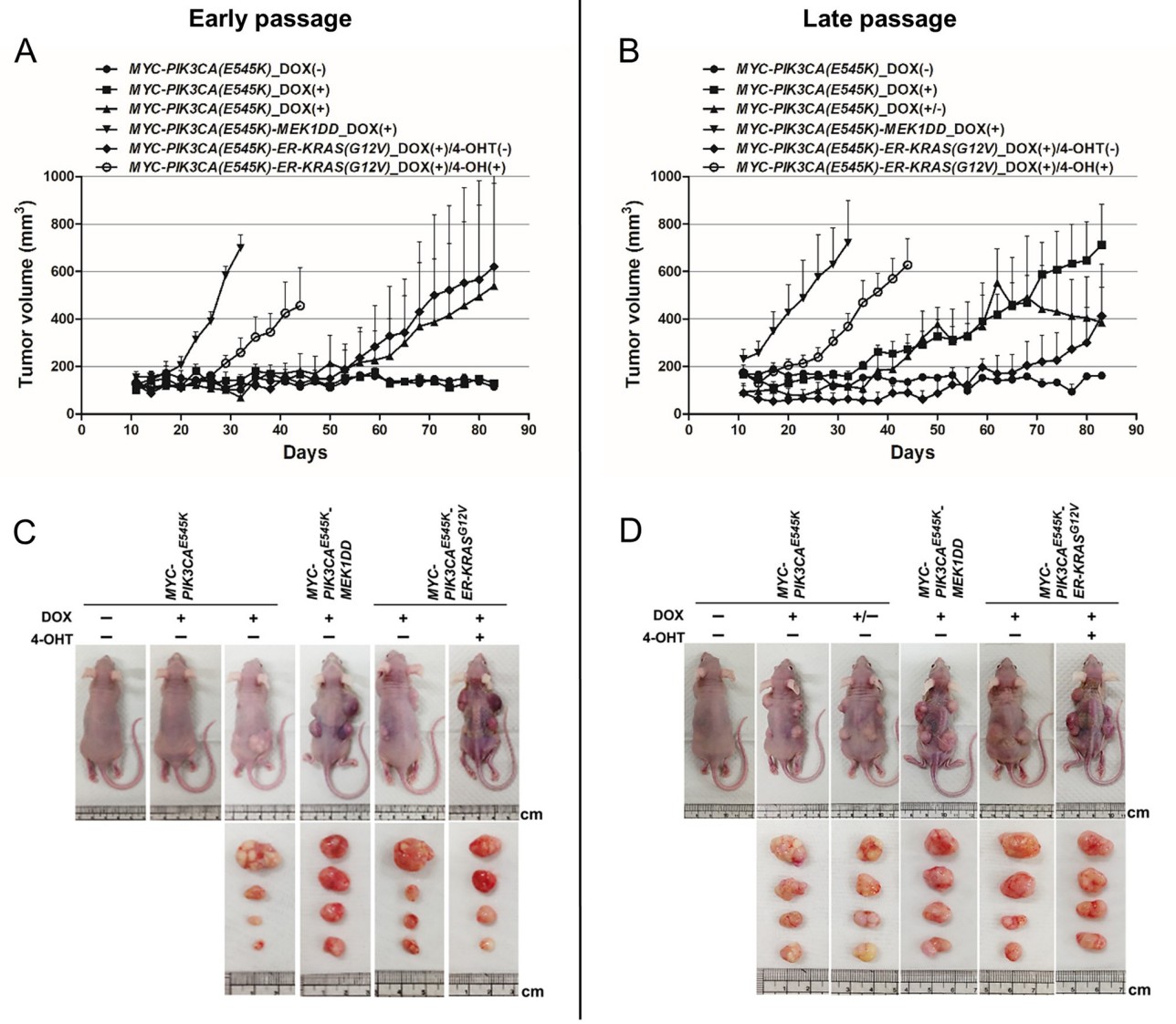

**Fig 4. The tumorigenicity of HCK1T/16epi cells expressing oncogenes in nude mice.** Tumor-promoting potentials of the *MYC, PIK3CA^E545K^, MEK1DD* and *ER-KRAS^G12V^* oncogenes were investigated by mouse xenografts. 1x10^6 cells mixed with Matrigel were subcutaneously injected into nude mice. Mice were given 1 mg/ml DOX and/or 0.2 mg/ml 4-OHT, or vehicle. The size of each tumor was measured at the indicated time points. The results from early-passage (**A**) and late-passage cells (**B**) are shown. Transgenic mice were sacrificed after tumor formation then tumor mass was compared for early-passage (**C**) and late-passage (**D**) HCK1T/16epi cells. DOX was removed from drinking water for mice injected with late-passage HCK1T/ HPV16epi cells with MYC and *PIK3CA^E545K^* after tumor size reached more than 400 mm^3.

## A Early passage

## B Late passage

**Fig 5. H&E staining of tumor tissues.** H&E staining of representative tumors isolated from nude mice injected with HCK1T/16epi expressing *MYC/PIK3CA^{E545K}* alone or with *MEK1DD* and/or *ER-KRAS^{G12V}*, at early passage (**A**) and at late passage (**B**). Scale bars represent 50 μm.

keratinocytes (NIKS) containing HPV16 episomal genomes showed a higher growth rate than LSIL-like lines and the parental NIKS line [32]. In addition, we also showed that ectopic expression of host oncogenes, including *MYC* and *PIK3CA*$^{E545K}$, could readily confer a tumorigenic phenotype to late-passage HCK1T/16epi but not to early-passage, suggesting that the late passage cells acquire susceptible traits for tumorigenic transformation. However, differences in biological behavior between early- and late-passage cells cannot be explained by expression levels of E6 and E7, since they are not elevated in late-passage HCK1T/16epi cells. These results were consistent with prior observations with a W12 cell line which contained only episomal HPV16 genomes established from a clinical specimen [33]. In the W12 model, subclones from a late passage exhibited increased ability to form colonies compared with the same clone from early passages, although expression levels of viral E2, E6 and E7 were somewhat higher in the cells at early passage. Further studies are needed to assess what alterations contribute to such phenotypic differences.

HR-HPV DNA integration into the host chromosome and subsequent loss of E2 protein inhibitory function on viral promoters might be the main cause of over-expression of E6 and E7 and a major step towards carcinogenesis [3–5]. While several studies have shown that integration occurs in most cancer samples [34,35] and cancer-derived cell lines [36], others have reported that some invasive cervical cancer specimens only contain episomal HPV genomes [8,9]. This implies that dysregulation of E6 and E7 expression can occur in the absence of integration. We have shown here that the HPV16 episomal genome can support cellular oncogene induced-tumorigenic transformation of HCK1T cells by enhancing proliferation and clonogenic potential. In our previous study, we found that diffuse strong p16INK4a staining was not correlated with HR-HPV integration. Episomal HR-HPV genomes were found in p16INK4a-positive squamous-cell carcinoma lesions [37]. In addition, no significant difference was seen in levels of E6 and E7 mRNA transcripts among cervical cancer samples whether they harbor only episomal HR-HPV or integrated HR-HPV [8,38]. This suggests that cancers containing episomal viral genomes may also have undergone transcriptional dysregulation to up-regulate E6/E7 mRNA expression, thus leading to disruption of the normal cell cycle, cell transformation and chromosomal instability.

Alterations of oncogenes such as *PIK3CA* and *MYC* are seen frequently in HPV-positive tumors, thus reflecting the close relationship between genetic aberrations and HPV infection [14,39]. Indeed, a recent piece of evidence indicated that the *PIK3CA*$^{E545K}$ mutation corresponds to the APOBEC signature [40] and that HPV infection induced activation of APOBEC [41,42]. *MYC* activation combined with HPV infection may be important for neuroendocrine cervical carcinogenesis [43]. *MYC* can cooperate with *RAS* to transform rodent cells [44], and we previously reported that the expression of *MYC* significantly enhances a tumor initiating property of HCKs expressing HPV16 *E6E7* driven from heterologous promoters and *HRAS*$^{G12V}$ [16]. Indeed, we found that significant co-occurrence of *MYC* amplification and *PIK3CA* alteration in TCGA of cervical cancer though the status of HPV genomes is not known. Consistently, ectopic expression of *MYC* and *PIK3CA*$^{E545K}$ with and without *MEK1DD* or *ER-KRAS*$^{G12V}$ enhances clonogenic potential of the late-passage HCK1T/16epi cells and hence tumor-forming ability. Furthermore, combined expression of *MYC/PIK3-CA*$^{E545K}$ and *MEK1DD* or *ER-KRAS*$^{G12V}$ in not only late-passage but also early-passage HCK1T/16epi cells gave rise to highly potent tumor-initiating cells (S1 Table). These results suggest overexpression of *MYC* and somatic mutations in *PIK3CA* such as *PIK3CA*$^{E545K}$ may facilitate neoplastic progression of HR-HPV episome containing cells in early-passage cells in this study. However, additional epigenetic and/or genetic alteration might be required for full transformation. In case of late-passage cells, a few somatic alterations such as MYC

overexpression and PIK3CA mutation may be sufficient to progress to tumors even without viral integration.

In summary, we have established a novel *in vitro* model for human cervical cancer harboring the episomal form of HPV16. Our results show that a few alterations of host oncogenes, such as *MYC*, *PIK3CA*, *MEK1* and *KRAS*, in conjunction with the episomal form of HPV16, might be sufficient to drive development of cervical cancer. Future use of this model will improve understanding of the roles of the episomal form of HPV and of specific oncogenes that are altered in cervical carcinogenesis.

## Supporting information

**S1 Fig. Southern blot hybridization for the HPV genome in HCK1T/16epi cells.** BamHI or EcoRV-digested total DNA isolated from HCK1T/16epi after transfection with PB-TA-C-ERN-MYC-F2A1-3xFLAG-PIK3CA$^{E545K}$ and CSII-TRE-Tight-HA-MEK1DD plasmid are shown. Digestion with BamHI, which cuts the HPV16 genome once, produced results of the expected size for the HPV16 genome. Digestion with EcoRV, which does not cut the HPV16 genome, showed open circular and supercoiled plasmid of HPV16 genome. The BamHI-linearized HPV16 plasmid was used for length and copy number standards. #1: HCK1T/HPV16epi (p35); #2: HCK1T/HPV16epi/MYC-PIK3CA$^{E454K}$/MEK1DD (p63); #3: HCK1T/HPV16epi/MYC-PIK3CA$^{E454K}$/MEK1DD (p47); #4: HCK1T-HPV16epi/MYC-PIK3CA$^{E454K}$ (p67).
(TIF)

**S2 Fig. The copy number of HPV16 genomes in tumors generated with the late-passage HCK1T/16epi expressing *MYC* and *PIK3CA*$^{E545K}$ was measured by qPCR using primers specific to the E6 and L2 ORFs of HPV16.** The bar represents the mean of triplicate values ± SEM.
(TIF)

**S3 Fig. Cell density-dependent growth of HCK1T/16epi late-passage cells expressing oncogenes were analyzed by clonogenic assays.** Each bar represents the mean of triplicate values ± SEM. $^{**}P \leq 0.01$, $^{***}P \leq 0.001$.
(TIF)

**S4 Fig.** The copy number of HPV16 genomes in tumor tissues of early-passage (A) and late-passage (B) cells were determined by qPCR using primers specific to the E6 and L2 ORFs of HPV16. Each bar represents the mean of triplicate values ± SEM.
(TIF)

**S1 Table. Numbers of nude mice developing tumors after injection of 1x10$^6$ HCK1T/16epi cells (per site) expressing *MYC*, *PIK3CA*$^{E545K}$, *MEK1DD* and *ER-KRAS*$^{G12V}$.**
(DOCX)

**S1 Raw images.**
(PDF)

## Acknowledgments

We would like to thank T. Ishiyama, Y. Yoshimatsu, C. Kohno, K. Dendo, E. Kabasawa and Y. Gotoh for expert technical assistance; Prof. David Blair for editing the MS via Publication Clinic KKU, Thailand.

## Author Contributions

**Conceptualization:** Weerayut Wongjampa, Tomomi Nakahara, Takashi Yugawa, Tohru Kiyono, Chamsai Pientong.

**Data curation:** Weerayut Wongjampa, Tomomi Nakahara, Tipaya Ekalaksananan, Tohru Kiyono, Chamsai Pientong.

**Formal analysis:** Weerayut Wongjampa, Tomomi Nakahara, Tohru Kiyono, Chamsai Pientong.

**Funding acquisition:** Weerayut Wongjampa, Pilaiwan Kleebkaow, Tohru Kiyono, Chamsai Pientong.

**Investigation:** Weerayut Wongjampa, Tomomi Nakahara, Tohru Kiyono, Chamsai Pientong.

**Methodology:** Weerayut Wongjampa, Tomomi Nakahara, Takashi Yugawa, Tipaya Ekalaksananan, Tohru Kiyono, Chamsai Pientong.

**Project administration:** Tomomi Nakahara, Tohru Kiyono, Chamsai Pientong.

**Resources:** Katsuyuki Tanaka, Takashi Yugawa, Pilaiwan Kleebkaow, Naoki Goshima.

**Supervision:** Tohru Kiyono, Chamsai Pientong.

**Writing – original draft:** Weerayut Wongjampa.

**Writing – review & editing:** Weerayut Wongjampa, Tomomi Nakahara, Tohru Kiyono, Chamsai Pientong.

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
