## [Decision Letter · Decision Letter 0]

5 Sep 2022

PONE-D-22-22148An in vitro carcinogenesis model for cervical cancer harboring episomal form of HPV16PLOS ONE

Dear Dr. 

Thank you for submitting your manuscript to PLOS ONE. After careful consideration, we feel that it has merit but does not fully meet PLOS ONE’s publication criteria as it currently stands. Therefore, we invite you to submit a revised version of the manuscript that addresses the points raised during the review process.

ACADEMIC EDITOR:

Dear Authors,

Thank you very much for submitting your manuscript to Plos One.

Our expert reviewers have commented on the manuscript. Please revise it according to their comments.

We look forward to your revision.

Best regards,

Plos One editorial office

We look forward to receiving your revised manuscript.

Kind regards,

Kazunori Nagasaka

Academic Editor

PLOS ONE

Additional Editor Comments :

Dear Authors,

Thank you very much for submitting your manuscript to Plos One.

Our expert reviewers have commented on the manuscript. Please revise it according to their comments.

We look forward to your revision.

Best regards,

Plos One editorial office

Reviewers' comments:

Reviewer's Responses to Questions

**Comments to the Author**

1. Is the manuscript technically sound, and do the data support the conclusions?

Reviewer #1: No

Reviewer #2: Yes

2. Has the statistical analysis been performed appropriately and rigorously? 

Reviewer #1: No

Reviewer #2: Yes

3. Have the authors made all data underlying the findings in their manuscript fully available?

Reviewer #1: No

Reviewer #2: Yes

4. Is the manuscript presented in an intelligible fashion and written in standard English?

Reviewer #1: Yes

Reviewer #2: Yes

5. Review Comments to the Author

Reviewer #1: In this study, the authors aim to investigate the question which additional factors are necessary to render cells tumorigenic that carry a purely episomal form of HPV16. They make use of a previously established cell system.

Major:

Especially the mouse experiments are not described in sufficient detail. How many mice per group were used? How many tumors were set per mouse? Do figures 2C and 4AB reflect the means of mice per group, or do they show single mice and show the mean of multiple tumors? There is also an inconsistency between the results text for Fig. 4, where 3 mice per group are mentioned for early passage MYC + PIK3CA/E545K mice, and Suppl. Table 4, where altogether 8 mice are stated for this treatment.

Most important: Throughout the Results section, the authors only describe part of the figures. However, the non-mentioned parts often contradict their claims. In order of importance:

1) Early passage cells with MYC and PIK3CA/E545K are not tumorigenic in Figure 2, and the authors continue to state that other factors are necessary. However, in Figure 4, the exact same cells are tumorigenic. Also the version of the cells with ER-KRAS but without the inducing agent are tumorigenic. It is thus not correct to conclude that additional factors are necessary, as the presented data show otherwise.

2) In Figure 1, the authors find that E6 and E7 levels are lower in late passage cells than in early passage cells. In Figure 2, it is the other way round, which is not discussed or even mentioned.

3) Figure 4 needs to be described much better. Why are there 2 groups with the same treatment in A – and why do they behave completely differently?

4) Figure 2, description of panels E and F: several proteins behave differently than in Fig. 2B.

The authors mention and describe “data not shown” in several parts of the manuscript. All described data should be shown. In the description of Figure 2 (lines 318ff) experiments are described that are not part of the figure.

The quality of some Western blot panels is not fit for publication (e.g. E6 in 2B). There obviously also is a loading issue in panels 2E and 2F, with clearly less protein in the last two or three slots, respectively.

The authors need to make sure to provide proper data, show and describe all data and methods, and adjust their conclusions so that they reflect the experimental results.

Minor:

Intro: The statements in line 64-66 should be supported by a reference.

Intro, line 93: Ref 11 could be replaced by a more well-known review of HPV biology.

Intro, line 133 ff: Not only the authors have shown previously that activated RAS is necessary to render E6E7-immortalized cells tumorigenic. This is actually well known in the field. The authors should cite the respective prior literature (e.g. the description of generation of the TC-1 cell line).

Methods, line 212: The methodology for the clonogenic assay seems rather short.

Statistics: The authors should double-check if standard error of the mean (SEM) is really the correct measure of variance for their data. Standard deviation (SD) seems more appropriate.

Results, lines 263 ff, referring to Fig. S1: It is not clear how copy number can be deducted from Fig. S1. The authors refer to subpopulations of cells with only 30 copy numbers, but no such populations are shown in the figure.

Results, line 297: MCM7 should be introduced. In the introduction, only Rb is mentioned as a E7 target protein. Furthermore, an increase is described, which cannot be seen in Fig 1.

Results, line 324: Also the upper panels of 2D should be mentioned, or otherwise removed. The histological pathological features for SCC should be enumerated, so that readers not used to histology can understand the shown staining (holds also true for Fig. 5).

Results, line 340: In the late passage cells, HPV copy numbers do go down markedly, contrary to the description.

Results, line 344: Describe more clearly which phosphorylated protein is a marker for functionality of which transgene. Early passage parental cells are not shown in Fig. 3E.

Results, line 365 and later: Nude mice are suddenly described as transgenic.

Results, lines 401-403: These data are not shown.

Discussion, line 464: When MYC and RAS are sufficient to transform rodent cells, the outhors should test if cells that have received both transgenes still need the HPV oncogene expression.

Discussion, lines 429 and 475: It is an overstatement that early passage HCK1T16epi cells resemble LSIL and late passage cells HSIL – no experiments to compare any biological features were done.

Editorial:

Results, lines 333/334: Make sure to stay with “DOX” – switching to “tet” without an explanation is confusing.

Results, Figure legend 3: The last sentence needs to be moved to the end of the description of panel C and D.

Discussion, first sentence: This sentence is incomplete.

Reviewer #2: The manuscript by Wongjampa et al. describes a novel cell culture system that recapitulates the tumorigenicity of HPV-induced cervical cancer, particularly one containing only the episomal form of HPV16. Starting from human cervical keratinocytes transduced with circular HPV16 genomes (HCK1T/16epi), the authors introduced various host oncogenes, MYC, PIK3CAE545K, MEK1DD, and KRASG12V, into the cells, and examined the tumor-forming ability of the resulting cells using mouse xenografts. Interestingly, forced expression of MYC and PIK3CAE545K conferred a tumorigenic phenotype to late passages of HCK1T/16epi, but not to their early passages. Moreover, additional introduction of MEK1DD or KRASG12V conferred tumorigenic potential also to the early passages of HCK1T/16epi.

Although the W12 cell line derived from HPV16-positive, CIN1 lesions has widely been used as a conventional system to analyze the life cycle and oncogenicity of HPV16, the cell culture system reported in this study not only provide a new in vitro model for only viral episome-containing cervical cancer, but also enable genetic manipulation of the HPV genome and cellular genes to define the roles of individual viral and host genes in cancer development, thus will contribute to a better understanding of HPV-induced carcinogenesis.

Major comments:

(1) To state that HPV integration is not always required for virus tumorigenicity, it is very important to show the presence of viral episomes during the course of cell culture. The data of virus southern blotting (Fig S1) should be presented with more time points and cell types.

(2) Line 272: “cell morphology and the viral copy numbers are stably maintained over at least 110 days or 50 population doublings.” It is interesting to see stable maintenance of HPV genomes for such a long period, but is it necessary to continuously add G418 to the culture medium? This should be clarified.

(3) Line 321: “The viral copy numbers measured by two independent primer sets were consistent” The data should be presented.

(4) Line 401: “In all tumors, HPV16 DNA was present mainly as an episomal form and the viral copy numbers were comparable with original 2D culture.” What does “mainly” mean? Is there any trace of an integration signal? Also, the data of viral copy numbers should be presented.

(5) The different behavior between early and late passages in inducing tumor growth in mouse is intriguing, but can this be fully reproduced with another batch of HCK1T/16epi cells? All experiments seem to be conducted with a pool of cells, but not a selected single clone?

(6) In Fig 3E and 3F, the expression levels of E7 seem to be lower in KRASG12V-transduced cells than other cells. “The expression levels of E6 and E7 proteins were comparable ~ ” (line 344-347) should be modified.

Minor comments:

(1) Caution should be taken regarding the paper by Hu et al. (ref. 10) because the integration frequency might be overestimated in that study due to unreliable identification of HPV integration reads by their bioinformatics analyses. Please see “Artifacts in the data of Hu et al” Nigel Dyer et al. Nat Genet. 2016 Jan;48(1):2-4.

(2) Line 423: “The molecular etiology of cervical cancer in which the episomal form of HR-HPV is poorly understood due to the lack of a well-defined model.” This sentence should be rephrased.

6. PLOS authors have the option to publish the peer review history of their article (what does this mean?). If published, this will include your full peer review and any attached files.

Reviewer #1: No

Reviewer #2: No

---

## [Author Response · Author response to Decision Letter 0]

19 Oct 2022

Response to the Academic Editor:

Journal Requirements:

Reply: We thank the editor for the suggestion. We have checked the manuscript style according to PLOS ONE's style requirements. Please see the revised manuscript.

Reply: We thank the editor for the suggestion. We have already included the details of mice anesthesia and sacrifice methods in the materials and methods section of the revised manuscript with track changes. Please see the materials and methods section in line 259-262. 

3. PLOS ONE now requires that authors provide the original uncropped and unadjusted images underlying all blot or gel results reported in a submission’s figures or Supporting Information files. This policy and the journal’s other requirements for blot/gel reporting and figure preparation are described in detail at https://journals.plos.org/plosone/s/figures#loc-blot-and-gel-reporting-requirements and https://journals.plos.org/plosone/s/figures#loc-preparing-figures-from-image-files.When you submit your revised manuscript, please ensure that your figures adhere fully to these guidelines and provide the original underlying images for all blot or gel data reported in your submission. See the following link for instructions on providing the original image data: https://journals.plos.org/plosone/s/figures#loc-original-images-for-blots-and-gels. In your cover letter, please note whether your blot/gel image data are in Supporting Information or posted at a public data repository, provide the repository URL if relevant, and provide specific details as to which raw blot/gel images, if any, are not available. Email us at plosone@plos.org if you have any questions.

Reply: We thank the editor for the comment and suggestion. We have already provided the raw blot images as a PDF file. Please see the supporting information file “S1_raw_images”.

4. We note that you have included the phrase “data not shown” in your manuscript. Unfortunately, this does not meet our data sharing requirements. PLOS does not permit references to inaccessible data. We require that authors provide all relevant data within the paper, Supporting Information files, or in an acceptable, public repository. Please add a citation to support this phrase or upload the data that corresponds with these findings to a stable repository (such as Fig share or Dryad) and provide and URLs, DOIs, or accession numbers that may be used to access these data. Or, if the data are not a core part of the research being presented in your study, we ask that you remove the phrase that refers to these data.

Reply: We thank the editor for the information. We have already added the reference to support the data and removed the phrase that refers to the data are not a core part of the research being present in our study. Please see the revised manuscript with track changes in line 262-265, 300-301 and 374.

Response to the Reviewer #1: 

In this study, the authors aim to investigate the question which additional factors are necessary to render cells tumorigenic that carry a purely episomal form of HPV16. They make use of a previously established cell system.

Major comments:

1. Especially the mouse experiments are not described in sufficient detail. How many mice per group were used? How many tumors were set per mouse? Do figures 2C and 4AB reflect the means of mice per group, or do they show single mice and show the mean of multiple tumors? There is also an inconsistency between the results text for Fig. 4, where 3 mice per group are mentioned for early passage MYC+PIK3CA/E545K mice, and Suppl. Table 4, where altogether 8 mice are stated for this treatment.

Reply: We thank the reviewer for the comment. 

In Figure 2C, the study of tumor-promoting potential of the MYC and PIK3CAE545K oncogenes that were compared between early and late passage of HCK1T/16epi cells by mouse xenografts. In this study, two mice were entirely used and four injection sites were set per mouse. The 1st mouse was injected with early passage cells containing MYC and PIK3CAE545K genes, and the 2nd mouse was injected with late passage cells containing MYC and PIK3CAE545K genes. 

We already added the description in the result section of the revised manuscript with track changes. Please see the result section in line 334-335.

In Figure 4, tumor-promoting potentials of MYC, PIK3CAE545K, MEK1DD and ER-KRASG12V oncogenes were investigated by mouse xenografts, a total of 12 mice were injected with early passage cells (6 mice) or late passage cells (6 mice) containing MYC, PIK3CAE545K, MEK1DD, and ER-KRASG12V genes, with DOX and 4-OHT activation and four injection site were set per mouse as shown in S1 Table. 

We already added the description in the result section of the revised manuscript with track changes. Please see the result section in line 409-410.

For the graphs representing tumor volume, each line represents a mouse and the dots on the curve at each time point represent the mean of the four positions of tumor mass in one mouse.

 In Figure 4A, the group injected with early passage cells containing MYC and PIK3CAE545K genes used a total of 3 mice, of which only 1 had tumors at all 4 sites, represented by numbers 4/4 as shown in S1 Table. 

In S1 Table, () showed latency that was determined as the time (weeks) taken before a palpable mass could be detected as noted. Therefore, numbers (8) or (3) in S1 Table represent the number of weeks in which the size of the tumor can be measured after DOX stimulation in HCK1T16epi/MYC-PIK3CAE545K and HCK1T16epi/MYC-PIK3CAE545K-MEK1DD, respectively.

2. Most important: Throughout the Results section, the authors only describe part of the figures. However, the non-mentioned parts often contradict their claims. In order of importance:

2.1 Early passage cells with MYC and PIK3CA/E545K are not tumorigenic in Figure 2, and the authors continue to state that other factors are necessary. However, in Figure 4, the exact same cells are tumorigenic. Also the version of the cells with ER-KRAS but without the inducing agent are tumorigenic. It is thus not correct to conclude that additional factors are necessary, as the presented data show otherwise.

Reply: We thank the reviewer for the comment. 

In Figure 2, one mouse was injected with early passage cells containing MYC and PIK3CAE545K genes, and no tumor was observed in this mouse. This study confirms the previous result. 

In Figure 4, a total of 3 mice were injected with early passage cells containing MYC and PIK3CAE545K genes. Two were DOX-stimulated via drinking water, but only one had tumors. In addition, the version of the early passage cells with MYC/PIK3CAE545K/ER-KRASG12V, a mouse was DOX-stimulated via drinking water to induce the expression of MYC and PIK3CA and this mouse also had tumors. The differences in these results may be due to the differences in the drinking behavior of the mice, resulting in different doses of DOX uptake and leading to different tumor formations.

 We already added the description in the result section of the revised manuscript with track changes. Please see the result section in line 420-422.

From Figure 4C, the tumors occurring in these groups of mice were very small and It took longer to develop tumors compared to those with the MEK1DD or ER-KRAS genes. Therefore, we conclude that the addition of the MEK1DD or ER-KRAS genes to the early passage cells is a key factor in the formation of tumors.

2.2 In Figure 1, the authors find that E6 and E7 levels are lower in late passage cells than in early passage cells. In Figure 2, it is the other way round, which is not discussed or even mentioned.

Reply: We thank the reviewer for the comment. In Figure 1 shows the expression levels of E6 and E7 in HCK1T/16epi cells, which were found to have higher expression levels in the early passage compared to the late passage. Contrary to Figure 2 shows the expression levels of E6 and E7 observed in HCK1T/16epi cells with the addition of MYC and PIK3CAE545K genes, which induce proliferation and clonogenicity of the cells and might be resulting in increased expression of E7 in the late passage.

2.3 Figure 4 needs to be described much better. Why are there 2 groups with the same treatment in A and why do they behave completely differently?

Reply: We thank the reviewer for the comment. In Figure 4A and 4C, a total of 3 mice were injected with early passage cells containing MYC and PIK3CAE545K genes. Two were DOX-stimulated via drinking water, but only one had tumors. The differences in results may be due to the differences in the drinking behavior of the mice, resulting in different doses of DOX uptake and leading to different tumor formations.

 We already added the description in the result section of the revised manuscript with track changes. Please see the result section in line 420-422. 

2.4 Figure 3, description of panels E and F: several proteins behave differently than in Fig. 2B.

Reply: We thank the reviewer for the comment. 

In Figures 3E and 3F, as the MEK1DD and ER-KRAS genes were added in addition to the MYC and PIK3CAE545K genes. Therefore, protein types associated with these genes were increased compared to Figure 2B. 

As you can see in the Figures 3E and 3F, increased phosphorylation of AKT and mTOR which are the downstream target of PIK3CAp100 and also ERK which is the downstream target of KRAS and MEK confirmed that induced transgenes are functionally active.

We have made corrections, please see the revised manuscript with track changes in line 363-365.

3. The authors mention and describe “data not shown” in several parts of the manuscript. All described data should be shown. In the description of Figure 2 (lines 318ff) experiments are described that are not part of the figure.

Reply: We thank the reviewer for the comment and apologize for the mistake. We have already added the reference to support the data and removed the phrase that refers to the data are not a core part of the research being present in our study. Please see the revised manuscript with track changes in line 262-265, 300-301 and 374.

For Figure 2, we have made corrections, please see the revised manuscript with track changes line 340-343 and included the figure in the supporting information: S2 Fig in S1 File.

4. The quality of some Western blot panels is not fit for publication (e.g. E6 in 2B). There obviously also is a loading issue in panels 2E and 2F, with clearly less protein in the last two or three slots, respectively.

Reply: We thank the reviewer for the comment. For Western blot, we are sure that there were no problems with protein loading, as observed with similar levels of the vinculin (control protein) across all wells. Because of the E6 protein has a very low expression. This makes the protein bar thin and unclear. However, decreased of p53 protein confirmed that E6 is expressed and functionally active. 

5. The authors need to make sure to provide proper data, show and describe all data and methods, and adjust their conclusions so that they reflect the experimental results.

Reply: We thank the reviewer for the comment. We make sure to provide accurate information, describe all data and methods such as in line 220-231 and also summarize the results in accordance with the experimental results. Please see the revised manuscript with track changes.

Minor comments:

1. Intro: The statements in line 64-66 should be supported by a reference.

Reply: We thank the reviewer for the suggestion. We have already added a reference according to the instructions. Please see the revised manuscript with track changes in line 66 and reference no. 7.

2. Intro, line 93: Ref 11 could be replaced by a more well-known review of HPV biology.

Reply: We thank the reviewer for the suggestion. We have already changed the reference according to the instructions. Please see the revised manuscript with track changes in line 96 and reference no. 11. 

3. Intro, line 133 ff: Not only the authors have shown previously that activated RAS is necessary to renderE6E7-immortalized cells tumorigenic. This is actually well known in the field. The authors should cite the respective prior literature (e.g. the description of generation of the TC-1 cell line).

Reply: We thank the reviewer for the suggestion. We have already added a reference according to the instructions. Please see the revised manuscript with track changes in line 116-119 and reference no. 23.

4. Methods, line 212: The methodology for the clonogenic assay seems rather short.

Reply: We thank the reviewer for the comment. We have already added the details of clonogenic assay in the materials and methods section of the revised manuscript with track changes. Please see the revised manuscript with track changes in line 220-231.

5. Statistics: The authors should double-check if standard error of the mean (SEM) is really the correct measure of variance for their data. Standard deviation (SD) seems more appropriate.

Reply: We thank the reviewer for the suggestion. We have checked and found that both methods are consistent. And in this experiment, we chose to use the standard error of the mean (SEM).

6. Results, lines 263 ff, referring to Fig. S1: It is not clear how copy number can be deducted from Fig. S1. The authors refer to subpopulations of cells with only 30 copy numbers, but no such populations are shown in the figure.

Reply: We thank the reviewer for the comment and apologize for the mistake. We have made corrections, please see the revised manuscript with track changes in line 280-283.

7. Results, line 297: MCM7 should be introduced. In the introduction, only Rb is mentioned as a E7 target protein. Furthermore, an increase is described, which cannot be seen in Fig 1.

Reply: We thank the reviewer for the suggestion. We have introduced the correlation between E7 and MCM7 in the results section of the revised manuscript with track changes. Please see the results section in line 315-316 and reference no. 31.

 In Figure 1E, it is seen that the level of MCM7 protein in early passage cells was slightly increased compared to parental cells, while those in late passage cells showed the same level as parental cells. This may be due to the low expression of E7 protein in the late passage cells.

8. Results, line 324: Also the upper panels of 2D should be mentioned, or otherwise removed. The histological pathological features for SCC should be enumerated, so that readers not used to histology can understand the shown staining (holds also true for Fig. 5).

Reply: We thank the reviewer for the suggestion. We have made corrections according to the instructions. Please see the revised manuscript with track changes in line 340-344.

9. Results, line 340: In the late passage cells, HPV copy numbers do go down markedly, contrary to the description.

Reply: We thank the reviewer for the comment. In figures 3C and 3D, the copy number of HPV16 DNA in late passage cells was slightly decreased when compared with early passage cells but did not show significant differences between the presence and the absence of the drugs (DOX and 4-OHT) in the same type of cells.

10. Results, line 344: Describe more clearly which phosphorylated protein is a marker for functionality of which transgene. Early passage parental cells are not shown in Fig. 3E.

Reply: We thank the reviewer for the suggestion. We have made corrections according to the instructions. Please see the revised manuscript with track changes in line 363-365.

 In Figure 3E, we apologize for the imperfection of the western blot experiment which we did not include early passage parental cells. However, we think that the western blot result that we present is sufficient to confirm that the transgenes are functionally active. 

11. Results, line 365 and later: Nude mice are suddenly described as transgenic.

Reply: We thank the reviewer for the comment. We have made corrections according to the instructions. Please see the revised manuscript with track changes in line 387.

12. Results, lines 401-403: These data are not shown.

Reply: We thank the reviewer for the comment and apologize for the mistake. We have made corrections, please see the revised manuscript with track changes in line 428-433 and included the figure in the supporting information: S4 Fig in S1 File.

13. Discussion, line 464: When MYC and RAS are sufficient to transform rodent cells, the authors should test if cells that have received both transgenes still need the HPV oncogene expression.

Reply: We thank the reviewer for the suggestion. We strongly agree with what you suggest.

14. Discussion, lines 429 and 475: It is an overstatement that early passage HCK1T16epi cells resemble LSIL and late passage cells HSIL – no experiments to compare any biological features were done.

Reply: We thank the reviewer for the comment and apologize for the mistake. We have made corrections, please see the revised manuscript with track changes in line 506-508. 

Response to the Editorial:

1. Results, lines 333/334: Make sure to stay with “DOX” – switching to “tet” without an explanation is confusing.

Reply: We thank the editor for the suggestion. We have made corrections according to the instructions. Please see the revised manuscript with track changes in line 353.

2. Results, Figure legend 3: The last sentence needs to be moved to the end of the description of panel C and D.

Reply: We thank the editor for the suggestion. We have made corrections according to the instructions. Please see the revised manuscript with track changes in line 398 and 403-404.

3. Discussion, first sentence: This sentence is incomplete.

Reply: We thank the editor for the comment. We have made corrections according to the instructions. Please see the revised manuscript with track changes in line 454-455.

Response to the Reviewer #2:

The manuscript by Wongjampa et al. describes a novel cell culture system that recapitulates the tumorigenicity of HPV-induced cervical cancer, particularly one containing only the episomal form of HPV16. Starting from human cervical keratinocytes transduced with circular HPV16genomes (HCK1T/16epi), the authors introduced various host oncogenes, MYC, PIK3CAE545K, MEK1DD, and KRASG12V, into the cells, and examined the tumor-forming ability of the resulting cells using mouse xenografts. Interestingly, forced expression of MYC and PIK3CAE545K conferred a tumorigenic phenotype to late passages of HCK1T/16epi, but not to their early passages. Moreover, additional introduction of MEK1DD or KRASG12V conferred tumorigenic potential also to the early passages of HCK1T/16epi.

Although the W12 cell line derived from HPV16-positive, CIN1 lesions has widely been used as a conventional system to analyze the life cycle and oncogenicity of HPV16, the cell culture system reported in this study not only provide a new in vitro model for only viral episome-containing cervical cancer, but also enable genetic manipulation of the HPV genome and cellular genes to define the roles of individual viral and host genes in cancer development, thus will contribute to a better understanding of HPV-induced carcinogenesis.

Major comments:

1. To state that HPV integration is not always required for virus tumorigenicity, it is very important to show the presence of viral episomes during the course of cell culture. The data of virus southern blotting (Fig S1) should be presented with more time points and cell types.

Reply: We thank the reviewer for the comment. In Figure S1, we examined the viral episomes in various passages to represent the cells used in the experiment. We actually tried to detect the viral episomes in tumors collected from mice, but the genomic DNA recovered from the tumors was not sufficient for southern blot analysis. However, we think that the information we provide is sufficient to confirm that the cells we use contain episomal HPV16 DNA.

2. Line 272: “cell morphology and the viral copy numbers are stably maintained over at least 110 days or 50 population doublings.” It is interesting to see stable maintenance of HPV genomes for such a long period, but is it necessary to continuously add G418 to the culture medium? This should be clarified.

Reply: We thank the reviewer for the comment. We have made corrections, please see the revised manuscript with track changes in line 137-138.

3. Line 321: “The viral copy numbers measured by two independent primer sets were consistent” The data should be presented.

Reply: We thank the reviewer for the comment and apologize for the mistake. We have made corrections, please see the revised manuscript with track changes in line 340-343 and included the figure in the supporting information: S2 Fig in S1 File.

4. Line 401: “In all tumors, HPV16 DNA was present mainly as an episomal form and the viral copy numbers were comparable with original 2D culture.” What does “mainly” mean? Is there any trace of an integration signal? Also, the data of viral copy numbers should be presented.

Reply: We thank the reviewer for the comment and apologize for the mistake. We have made corrections, please see the revised manuscript with track changes in line 428-433 and included the figure in the supporting information: S4 Fig in S1 File.

5. The different behavior between early and late passages in inducing tumor growth in mouse is intriguing, but can this be fully reproduced with another batch of HCK1T/16epi cells? All experiments seem to be conducted with a pool of cells, but not a selected single clone?

Reply: We thank the reviewer for the comment. We are confident that the experiment can be repeated. This is because we have stocked the HCK1T16epi cells in the early passages so when we want to perform the experiment, we can use the same passage of cells with the previous study.

6. In Fig 3E and 3F, the expression levels of E7 seem to be lower in KRASG12V-transduced cells than other cells. “The expression levels of E6 and E7 proteins were comparable ~ ” (line 344-347) should be

Reply: We thank the reviewer for the comment. We have made corrections according to the instructions. Please see the revised manuscript with track changes in line 366-369.

Minor comments:

1. Caution should be taken regarding the paper by Hu et al. (ref. 10) because the integration frequency might be overestimated in that study due to unreliable identification of HPV integration reads by their bioinformatics analyses. Please see “Artifacts in the data of Hu et al” Nigel Dyer et al. Nat Genet. 2016 Jan;48(1):2-4.

Reply: We thank the reviewer for the suggestion. We have already changed the reference according to the instructions. Please see the revised manuscript with track changes in line 75-89 and reference no. 10.

2. Line 423: “The molecular etiology of cervical cancer in which the episomal form of HR-HPV is poorly understood due to the lack of a well-defined model.” This sentence should be rephrased.

Reply: We thank the reviewer for the comment. We have made corrections according to the instructions. Please see the revised manuscript with track changes in line 454-455.

---

## [Decision Letter · Decision Letter 1]

28 Nov 2022

PONE-D-22-22148R1An in vitro carcinogenesis model for cervical cancer harboring episomal form of HPV16PLOS ONE

Dear Dr. Pientong,

Thank you for submitting your manuscript to PLOS ONE. After careful consideration, we feel that it has merit but does not fully meet PLOS ONE’s publication criteria as it currently stands. Therefore, we invite you to submit a revised version of the manuscript that addresses the points raised during the review process.

ACADEMIC EDITOR:  Please revise the experimental results and, if necessary, send us your rebuttal letter.

We look forward to receiving your revised manuscript.

Kind regards,

Kazunori Nagasaka

Academic Editor

PLOS ONE

Additional Editor Comments (if provided):

Dear Authors,

Thank you very much for submitting your revised manuscript. Although most of the concerns have been corrected and the necessary experiments have been performed, please submit a rebuttal letter following the reviewers' comments.

Sincerely,

Plos one

Reviewers' comments:

Reviewer's Responses to Questions

**Comments to the Author**

1. If the authors have adequately addressed your comments raised in a previous round of review and you feel that this manuscript is now acceptable for publication, you may indicate that here to bypass the “Comments to the Author” section, enter your conflict of interest statement in the “Confidential to Editor” section, and submit your "Accept" recommendation.

Reviewer #1: (No Response)

Reviewer #3: All comments have been addressed

2. Is the manuscript technically sound, and do the data support the conclusions?

Reviewer #1: No

Reviewer #3: Yes

3. Has the statistical analysis been performed appropriately and rigorously? 

Reviewer #1: No

Reviewer #3: Yes

4. Have the authors made all data underlying the findings in their manuscript fully available?

Reviewer #1: Yes

Reviewer #3: Yes

5. Is the manuscript presented in an intelligible fashion and written in standard English?

Reviewer #1: Yes

Reviewer #3: Yes

6. Review Comments to the Author

Reviewer #1: The authors replied to all points raised, but did not perform a single additional experiment.

Most importantly, their answers confirmed that, indeed, not groups of mice were used, but a single mouse per condition for figures 2 and 4! The error bars just stem from the fact that 4 tumors were set per mouse.

It is completely unacceptable to base any conclusions on experiments with a single animal per condition. It is now clear why there are such large differences between the exact same conditions between Figure 2 and Figure 4, or within Figure 4. As these are not groups of mice, but single animals, differences are to be expected. For example because single mice can have different drinking behaviours, and thus take in less DOX or 4-OHT. This is exactly the reason why there have to be several mice in a group.

A proper study should involve a biostatistician before the experiment is conducted, an estimation of the size of the effect to be tested (and the required statistical significance), and then a calculation of the necessary number of mice per experimental group.

The authors also did not get my point that groups (or single mice) that did not receive the transgene-inducing agent should behave like the groups without the transgene. They continue just to describe the findings that fit their hypothesis, and not the findings (shown in figures) that do not. E.g. in panels 3E and 3F.

I reiterate that the E6 blot in Figure 2B is not fit for publication, and that there is a loading problem in Figures 3E (last two lanes) and 3F (last three lanes), which can also be seen in the Vinculin staining. It is not sufficient just to state that there is no problem, the experiments should be repeated until proper blots can be shown.

Regarding the suggested experiment, to test if cells that contain activated MYC and RAS – described as sufficient to transform rodent cells in the discussion – still need HPV oncogenes at all, the authors just wrote “We strongly agree with what you suggest”. As they agree, they really need to perform this experiment!

Reviewer #3: The authors in this manuscript described the full transformation of HCK1T cell line, carrying episomal HPV16 genomes, following ectopic expression of MYC and PIK3CAE545K. Overall the manuscript is well written and study results well presented

7. PLOS authors have the option to publish the peer review history of their article (what does this mean?). If published, this will include your full peer review and any attached files.

Reviewer #1: No

Reviewer #3: No

---

## [Author Response · Author response to Decision Letter 1]

10 Jan 2023

Response to the Reviewer #1:

Reviewer #1: The authors replied to all points raised, but did not perform a single additional experiment.

• Most importantly, their answers confirmed that, indeed, not groups of mice were used, but a single mouse per condition for figures 2 and 4! The error bars just stem from the fact that 4 tumors were set per mouse.

• It is completely unacceptable to base any conclusions on experiments with a single animal per condition. It is now clear why there are such large differences between the exact same conditions between Figure 2 and Figure 4, or within Figure 4. As these are not groups of mice, but single animals, differences are to be expected. For example because single mice can have different drinking behaviours, and thus take in less DOX or 4-OHT. This is exactly the reason why there have to be several mice in a group.

• A proper study should involve a biostatistician before the experiment is conducted, an estimation of the size of the effect to be tested (and the required statistical significance), and then a calculation of the necessary number of mice per experimental group.

Reply: Firstly, we would like to thank you for the comment, and regarding the number of mice that we used in this experiment; we would like to explain that our research plan to minimize the use of mice in order to comply with the research and achieve the objectives of the study. However, we are aware of important biological effects that can be lost if too few mice are used. We also looked at previous studies on the number of mice used in experiments similar to this study. It was found that one mouse per condition was used and four or six sites per mouse were also cells injected (Narisawa-Saito et. al., 2008, Narisawa-Saito et. al., 2012). 

For the advantage of using one mouse per condition is that it minimizes the genetic variation and environmental effects that can be occurred if multiple mice are used. But the advantage of using multiple mice per condition, as you suggest, can reduce the impact that might have on experiments involving different animal behavior, such as different drinking habits. 

However, even if there are factors related to animal behavior outside of the control as mentioned and affecting the experiment. In this experiment, we used mice as a suitable study and have followed the guidelines for the proper use of animals in scientific work.

We sincerely apologize for not being able to repeat the experiment or do more experiments as you suggest. Because our research group has changed its approach of studying and some members have changed jobs to a new organization. However, the research presented here is properly confirmed by our research group and considered to be suitable for publication and dissemination.

References

 Narisawa-Saito M, Yoshimatsu Y, Ohno S-i, Yugawa T, Egawa N, Fujita M, et al. An in vitro multistep carcinogenesis model for human cervical cancer. Cancer research. 2008;68(14):5699-705.

Narisawa-Saito M, Inagawa Y, Yoshimatsu Y, Haga K, Tanaka K, Egawa N, et al. A critical role of MYC for transformation of human cells by HPV16 E6E7 and oncogenic HRAS. Carcinogenesis. 2012;33(4):910-7.

• The authors also did not get my point that groups (or single mice) that did not receive the transgene-inducing agent should behave like the groups without the transgene. They continue just to describe the findings that fit their hypothesis, and not the findings (shown in figures) that do not. E.g. in panels 3E and 3F.

Reply: We thank the reviewer for the comment. 

First of all, we would like to apologize for the previous answer that did not understand the point you wish to discuss. However, we agree with you on the point that mice without transgene inducers should have the same properties as mice without transgene. As shown in Figure 4, mice that were not received transgene inducers did not develop tumors. In addition, mice given the transgene inducer develop tumors early. But when the doping was stopped, the tumor size gradually decreased.

In figures 3E and 3F, protein expression was examined by western blot in HCK1T/16epi cells treated with transgene inducers to confirm the function of those transgenes in comparison with HCK1T parental cells. As you can see in Figures 3E and 3F, increased phosphorylation of AKT and mTOR which are the downstream target of PIK3CA, and also ERK which is the downstream target of KRAS and MEK confirmed that induced transgenes are functionally active. 

However, from Figure 3F, it can be seen that in HCK1T/16epi with MYC-PIK3CAE545K-ER-KRASG12V cells that did not expose to transgene inducers had slightly higher expression levels of MEK and ERK proteins when compared with HCK1T parental cells. This result might be the effect from the leakage expression of transgenes that are commonly occurred in the constructed vectors and may affect on downstream target protein activation.

• I reiterate that the E6 blot in Figure 2B is not fit for publication, and that there is a loading problem in Figures 3E (last two lanes) and 3F (last three lanes), which can also be seen in the Vinculin staining. It is not sufficient just to state that there is no problem, the experiments should be repeated until proper blots can be shown.

Reply: We thank the reviewer for the comment and suggestion. 

First of all, we would like to apologize for not being able to repeat the experiments as you suggest and sincerely apologize for the previous comment regarding the problem to detect E6 protein expression by western blot. According to vinculin staining, we found that the E6 protein expression was detected at low levels when compared with E6 gene expression level in Figure1C and the protein band is unclear. This problem might be effected from protein loading and the antibody clone used. However, we try to make it more clear as shown in Figure 2B and E6 gene expression can be confirmed by the level of mRNA detected in the cells, shown in Figure 1C. 

We already added the description in the result section of the revised manuscript with track changes. Please see the result section in line 319-322.

• Regarding the suggested experiment, to test if cells that contain activated MYC and RAS – described as sufficient to transform rodent cells in the discussion – still need HPV oncogenes at all, the authors just wrote “We strongly agree with what you suggest”. As they agree, they really need to perform this experiment!

Reply: We thank the reviewer for the comment and suggestion. We think that your hypothesis is an interesting point and it is necessary to explore further with many experiments. However, we sincerely apologize for not being able to do more experiments as you suggest. As we already tell you that our research group has changed its approach of studying and some members have changed jobs to a new organization. As a result, we were unable to perform this experiment. 

Response to the Reviewer #3:

Reviewer #3: The authors in this manuscript described the full transformation of HCK1T cell line, carrying episomal HPV16 genomes, following ectopic expression of MYC and PIK3CAE545K. Overall the manuscript is well written and study results well presented

Reply: We thank the reviewer for the comment.

---

## [Editor Report · Decision Letter 2]

16 Jan 2023

An in vitro carcinogenesis model for cervical cancer harboring episomal form of HPV16

PONE-D-22-22148R2

Dear Dr. Pientong,

We’re pleased to inform you that your manuscript has been judged scientifically suitable for publication and will be formally accepted for publication once it meets all outstanding technical requirements.

Kind regards,

Kazunori Nagasaka

Academic Editor

PLOS ONE

Additional Editor Comments (optional):

Dear Authors,

The authors have researched thoroughly and sufficiently throughout, and the hypotheses have been adequately tested. The reviewers' questions are also answered.
---

## [Editor Report · Acceptance letter]

2 Feb 2023

PONE-D-22-22148R2 

An *in vitro* carcinogenesis model for cervical cancer harboring episomal form of HPV16 

Dear Dr. Pientong:

I'm pleased to inform you that your manuscript has been deemed suitable for publication in PLOS ONE. Congratulations! Your manuscript is now with our production department. 

Kind regards, 

on behalf of

Professor Kazunori Nagasaka 

Academic Editor

PLOS ONE